# Specific labeling of synaptic schwann cells reveals unique cellular and molecular features

Ryan Castro[1,2,3], Thomas Taetzsch[1,2], Sydney K Vaughan[1,2], Kerilyn Godbe[4], John Chappell[4], Robert E Settlage[5], Gregorio Valdez[1,2,6]*

[1]Department of Molecular Biology, Cellular Biology, and Biochemistry, Brown University, Providence, United States; [2]Center for Translational Neuroscience, Robert J. and Nancy D. Carney Institute for Brain Science and Brown Institute for Translational Science, Brown University, Providence, United States; [3]Neuroscience Graduate Program, Brown University, Providence, United States; [4]Fralin Biomedical Research Institute at Virginia Tech Carilion, Roanoke, United States; [5]Department of Advanced Research Computing, Virginia Tech, Blacksburg, United States; [6]Department of Neurology, Warren Alpert Medical School of Brown University, Providence, United States

**Abstract** Perisynaptic Schwann cells (PSCs) are specialized, non-myelinating, synaptic glia of the neuromuscular junction (NMJ), that participate in synapse development, function, maintenance, and repair. The study of PSCs has relied on an anatomy-based approach, as the identities of cell-specific PSC molecular markers have remained elusive. This limited approach has precluded our ability to isolate and genetically manipulate PSCs in a cell specific manner. We have identified neuron-glia antigen 2 (NG2) as a unique molecular marker of S100β+ PSCs in skeletal muscle. NG2 is expressed in Schwann cells already associated with the NMJ, indicating that it is a marker of differentiated PSCs. Using a newly generated transgenic mouse in which PSCs are specifically labeled, we show that PSCs have a unique molecular signature that includes genes known to play critical roles in PSCs and synapses. These findings will serve as a springboard for revealing drivers of PSC differentiation and function.

*For correspondence:
gregorio_valdez@brown.edu

**Competing interests:** The authors declare that no competing interests exist.

## Introduction

The neuromuscular junction (NMJ) is a tripartite synapse comprised of an α-motor neuron (the presynapse), extrafusal muscle fiber (the postsynapse), and specialized synaptic glia called perisynaptic Schwann cells (PSCs) or terminal Schwann cells. Due to its large size and accessibility, extensive research of the NMJ has been essential to the discovery of the fundamental mechanisms that govern synaptic function, including the concepts of neurotransmitter release, quantal transmission, and active zones, among others (*Katz and Miledi, 1967*; *Fatt and Katz, 1952*; *Sealock et al., 1989*; *Sobel et al., 1979*; *Sobel et al., 1977*; *Sanes and Lichtman, 1999*; *Darabid et al., 2014*; *Katz and Miledi, 1966*; *Robertson, 1956*; *Changeux et al., 1970*; *Godfrey et al., 1984*; *Jennings et al., 1993*; *Lwebuga-Mukasa et al., 1976*; *Nitkin et al., 1987*; *Porter and Froehner, 1983*). Likewise, the concept of glia that exist primarily to support synapse function, and thus the realization that synapses are tripartite, has its origins at the NMJ (*Robertson, 1956*; *Couteaux, 1960*; *Kang et al., 2007*; *Zuo et al., 2004*; *Griffin and Thompson, 2008*; *Boeke, 1949*; *Heuser and Reese, 1973*; *Miledi and Slater, 1968*; *Miledi and Slater, 1970*; *Peper et al., 1974*; *Astrow et al., 1994*; *Astrow et al., 1998*; *Reynolds and Woolf, 1992*; *Young et al., 2005*). PSCs surround the NMJ

where they are closely associated with its pre- and postsynaptic components (*Griffin and Thompson, 2008*; *Ko and Robitaille, 2015*; *Darabid et al., 2014*). In addition to providing trophic support for the NMJ (*Griffin and Thompson, 2008*; *Ko and Robitaille, 2015*; *Darabid et al., 2014*; *Reddy et al., 2003*), PSCs have been shown to guide motor axon innervation and synaptogenesis (*Reddy et al., 2003*; *Trachtenberg and Thompson, 1997*; *Koirala et al., 2000*; *O'Malley et al., 1999*; *Barik et al., 2016*), support compensatory axonal sprouting (*Astrow et al., 1994*; *Reynolds and Woolf, 1992*; *Son and Thompson, 1995*; *Love and Thompson, 1998*), participate in synaptic pruning (*Griffin and Thompson, 2008*; *Lee et al., 2017*; *Smith et al., 2013*; *Darabid et al., 2013*), and detect and modulate cholinergic transmission (*Ko and Robitaille, 2015*; *Jahromi et al., 1992*; *Reist and Smith, 1992*; *Robitaille, 1995*; *Robitaille et al., 1997*; *Rochon et al., 2001*).

While great progress has been made in understanding the cellular and physiological characteristics of PSCs, very little is known about the molecular composition of these cells (*Ko and Robitaille, 2015*). This has been due to the absence of a cell-specific molecular marker with which PSCs can be identified, isolated, and genetically manipulated. This has hindered examinations of the processes of PSC development, differentiation and turnover. Additionally, isolation and targeting of PSCs for interrogation of molecular function in vivo and in vitro has not been possible. Therefore, the discovery of markers specific to PSCs is necessary to advance our understanding of PSCs, and synaptic glia in general, on multiple fronts.

A growing number of molecular markers that recognize subsets of glial cells throughout the nervous system have been identified (*Jäkel and Dimou, 2017*). Therefore, we explored the possibility that a unique combination of glial cell markers could be used to distinguish PSCs. We have found that PSCs can be identified by the combined expression of the calcium-binding protein B (S100β) (*Brockes et al., 1979*; *Perez and Moore, 1968*) and neuron-glia antigen-2 (NG2) (*Stallcup, 1981*; *Bergles et al., 2010*) genes. We utilized this unique molecular fingerprint to create a transgenic mouse that enables visualization and isolation of PSCs in a cell specific manner. This genetic model will help overcome obstacles to understanding the cellular and molecular rules that govern PSC function at NMJs during development, following injury, in old age, and in diseases, such as Amyotrophic Lateral Sclerosis (ALS).

## Results

To identify unique markers for PSCs, we examined the expression of genes shown to be co-expressed with well-established markers of Schwann cells in a subset of glial cells in the central nervous system in PSCs. We focused on NG2 for the following reasons: 1) it has been found to be co-expressed with S100β, a classical marker of all Schwann cells, in a subset of glial cells in the developing brain (*Matthias et al., 2003*; *Hachem et al., 2005*; *Vives et al., 2003*; *Moshrefi-Ravasdjani et al., 2017*; *Platel et al., 2009*); 2) published data show that it is expressed in skeletal muscles and labels pericytes and neural progenitor cells (*Birbrair et al., 2013a*; *Birbrair et al., 2013b*). To determine if NG2 is expressed by PSCs, we examined whole-mounted extensor digitorum longus (EDL) muscles from NG2-dsRed mice (*Zhu et al., 2008*). We observed widespread distribution of NG2-dsRed postive cells in the EDL muscle, including a distinct subset of cells located specifically at the NMJ and with a similar morphology as PSCs (*Figure 1C*). To determine if NG2 is expressed in PSCs we generated a transgenic mouse line (referred herein as S100β-GFP;NG2-dsRed; *Figure 1A*) by crossing the NG2-dsRed line with the S100β-GFP mouse line in which the S100β promoter drives expression of GFP in all Schwann cells (*Zuo et al., 2004*). As expected, in the resulting S100β-GFP;NG2-dsRed double transgenic mouse line, dsRed labeled all NG2 positive cells (referred herein as NG2-dsRed[+]) and GFP labeled all Schwann cells (referred herein as S100β-GFP[+]) (*Figure 1B–C*) in skeletal muscles. However, we found a select subset of glia positive for both S100β-GFP and NG2-dsRed specifically located at the NMJ (yellow cells in *Figure 1D*). Based on the location and morphology of the cell body and its elaborations, we concluded that PSCs are the only cells expressing both S100β-GFP and NG2-dsRed in skeletal muscles.

We next evaluated whether the S100β-GFP;NG2-dsRed mouse line serves as a reliable model to study PSCs and their roles at NMJs. In healthy young adult muscle, we observed the same number of PSCs at NMJs in the EDL muscle of S100β-GFP and S100β-GFP;NG2-dsRed mice (*Figure 2A–C*). The morphology of PSCs also appeared to be indistinguishable between the two transgenic lines. In addition, the morphology of NMJs, as assessed by fragmentation of nicotinic acetylcholine receptor

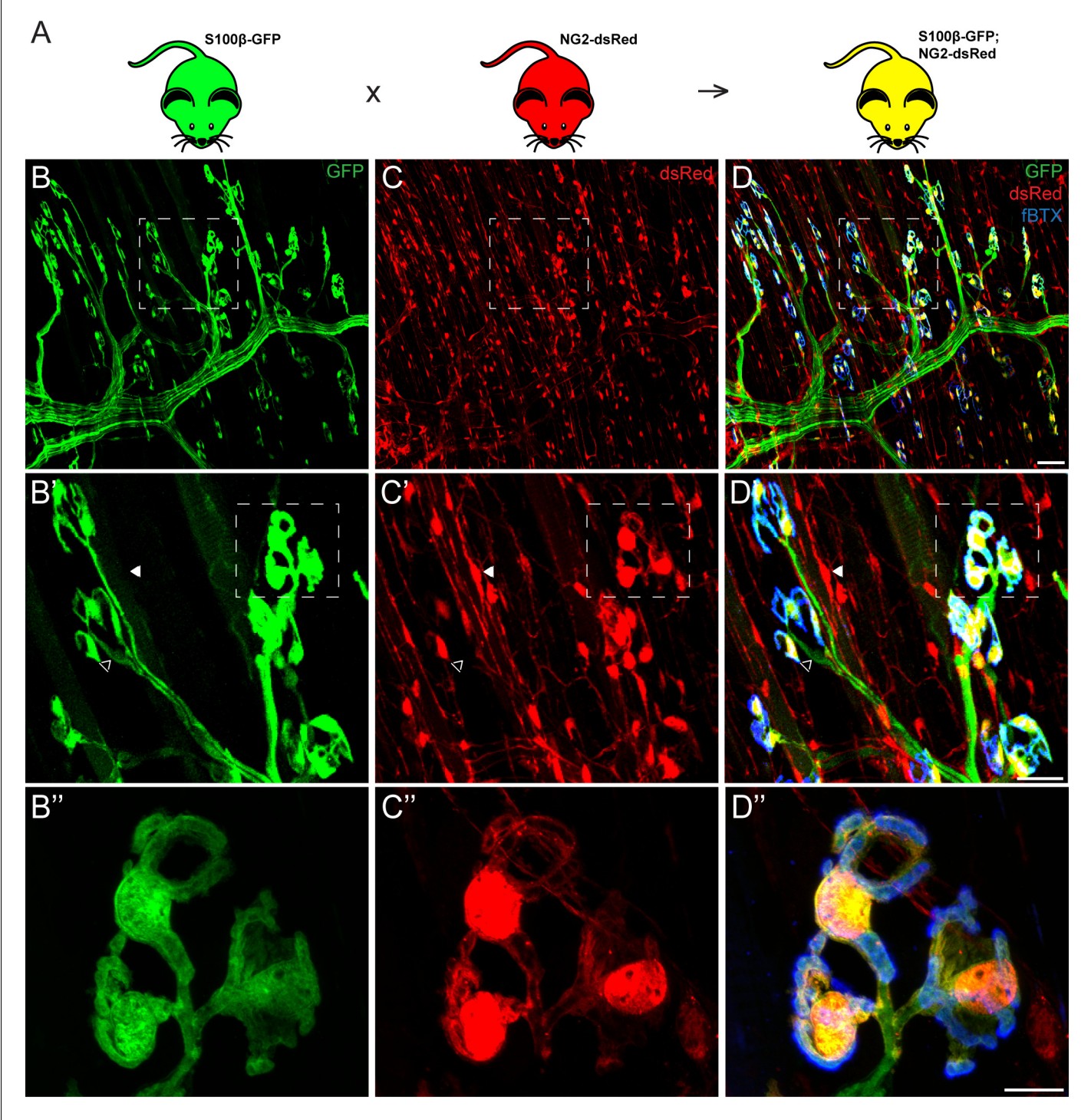

**Figure 1.** Co-expression of S100β and NG2 is unique to PSCs in the EDL muscle. (A) In order to selectively label PSCs, S100β-GFP and NG2-dsRed transgenic mice were crossed to create S100β-GFP;NG2-dsRed mice. (B–D) Representative images of GFP (B) and dsRed (C) fluorescence in the EDL of S100B-GFP;NG2-dsRed mice. S100β-GFP+ Schwann cells are visible along the motor axon while S100β-GFP+ PSCs are identified by their unique morphology and clustering pattern near the NMJ, visualized here using a fluourescent α-bungarotoxin conjugate (fBTX) to detect nAChRs (blue). Note that PSCs are the only cells expressing both GFP and dsRed (D). At non-synaptic sites, GFP-positive cells do not express dsRed (hollow arrow; B', C', D') and dsRed-positive cells do not express GFP (filled arrow; B', C', D'). Scale bar = 50 μm (D), 25 μm (D'), and 10 μm (D'').

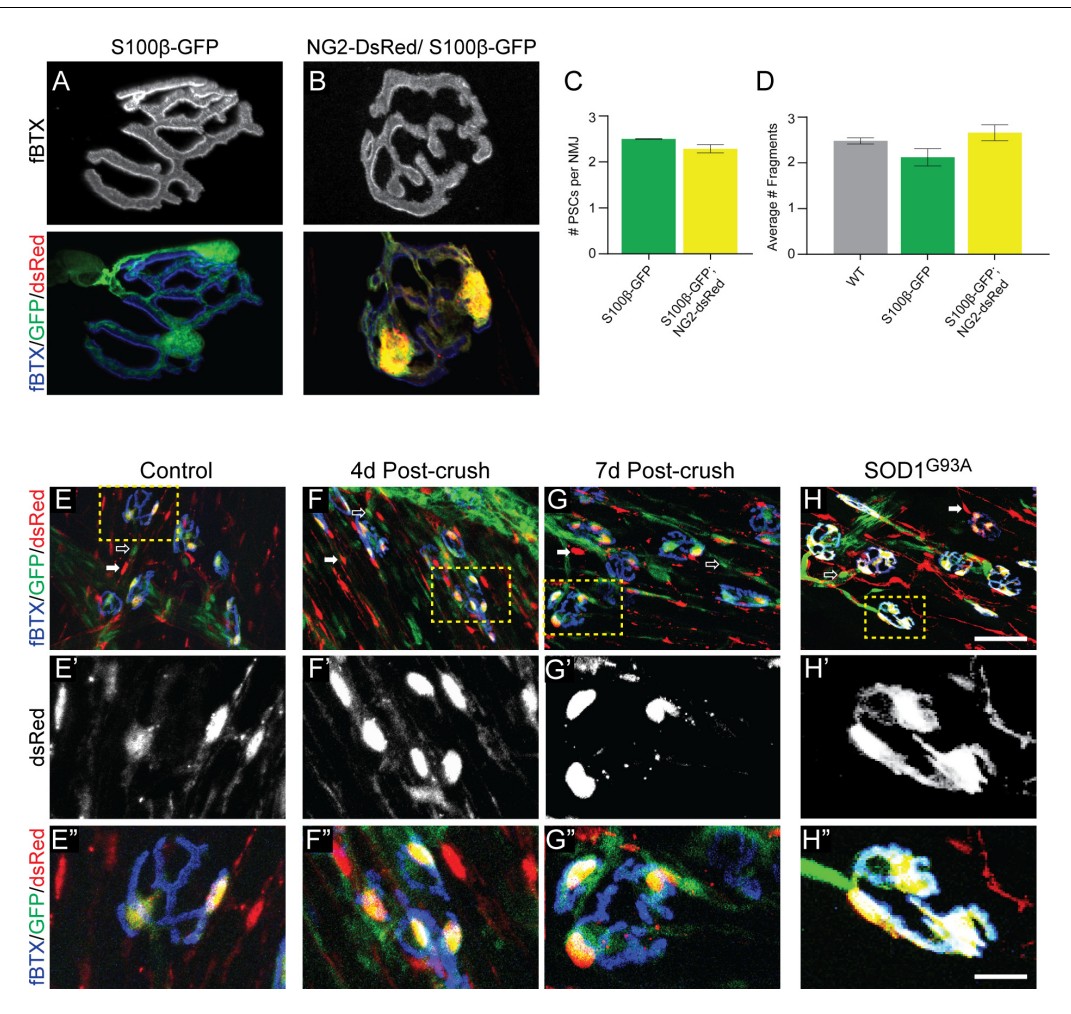

**Figure 2.** The NG2-DsRed/S100β-GFP mouse line can be used to reliably identify PSCs in healthy and stressed NMJs. (A–B) Representative images of NMJs identified by fBTX labeled nAChRs from S100β-GFP (A) and S100β-GFP;NG2-dsRed (B) EDL. (C–D) The co-expression of GFP and dsRed has no discernible negative effects on NMJ fragmentation or PSC number in the EDL muscle of young adult mice. (C) The average number of PSCs per NMJ is unchanged between S100B-GFP mice and S100β-GFP;NG2-dsRed mice. (D) The average number of nAChR fragments per NMJ, as determined by analysis of continuity of fBTX labeled nAChRs, is unchanged between wild-type, S100B-GFP, and S100β-GFP;NG2-dsRed animals. (E–H) PSCs in stressed muscle co-express S100β-GFP and NG2-dsRed. Representative images of NMJs identified by fBTX labeled nAChRs in S100β-GFP;NG2-dsRed mice shows co-expression of S100β-GFP and NG2-dsRed by PSCs in healthy uninjured (E), at 4d (F), and 7d (G) post-fibular nerve crush, and in P120 SOD1G93A (H) EDL. At non-synaptic sites, GFP-positive cells do not express dsRed (hollow arrow; E, F, G, H) and dsRed-positive cells do not express GFP (filled arrow; E, F, G, H). Error bar = standard error of the mean. Scale bar = 10 μm (A–B), 50 μm (E–H), and 12.5 μm (E'-H' and E'=H').

(nAChR) clusters, is unchanged in S100β-GFP;NG2-dsRed mice compared to S100β-GFP and wild type mice (*Figure 2A,B,D*). Thus, the co-expression of S100β-GFP and NG2-dsRed does not appear to cause apparent deleterious changes on either PSCs or the postsynaptic region revealed by nAChRs. However, it remains possible that co-expression of these markers in PSCs may disrupt the presynapse and biophysical properties of the NMJ. If so, we hypothesize that such changes would be minor given that S100β-GFP;NG2-dsRed mice are outwardly indistinguishable when compared to S100β-GFP and wild type mice.

We next assessed if S100β-GFP;NG2-dsRed mice could also be used to study PSCs at degenerating and regenerating NMJs. First, we examined expression of NG2-dsRed and S100β-GFP after crushing the fibular nerve (*Dalkin et al., 2016*). In this injury model, motor axons completely retract within 1 day and return to reinnervate vacated postsynaptic sites by 7 days post-injury in young adult mice. Similar to healthy uninjured EDL muscles (*Figure 2E*), NG2-dsRed and S100β-GFP were found co-expressed exclusively in PSCs at 4d and 7d post-injury (*Figure 2F–G*). Second, we crossed the

SOD1[G93A] mouse line, (*Gurney et al., 1994*) a model of ALS shown to exhibit significant degeneration of NMJs (*Moloney et al., 2014*), with S100β-GFP;NG2-dsRed mice and examined the expression pattern of NG2-dsRed and S100β-GFP in the EDL during the symptomatic stage (P120). We again found NG2-dsRed and S100β-GFP co-expressed only in PSCs in the EDL of P120 SOD1[G93A]; S100β-GFP;NG2-dsRed mice (*Figure 2H*). Together, these data strongly indicate that this genetic labeling approach can be deployed to study the synaptic glia of the NMJ in a manner previously not possible in healthy and stressed NMJs.

To determine the relationship between NG2 expression and PSC differentiation, we analyzed NG2 expression in S100β-GFP+ Schwann cells during the course of NMJ development in the EDL muscle of S100β-GFP;NG2-dsRed mice (*Figure 3A* and *Figure 3—figure supplement 1*). We observed the presence of S100β-GFP+ cells at the NMJ as early as embryonic day 15 (E15) with 100% of NMJs having at least one S100β-GFP+ cell by post-natal day 9 (*Figure 3A,B*). During the embryonic developmental stages, we observed that NMJs are exclusively populated by S100β-GFP+

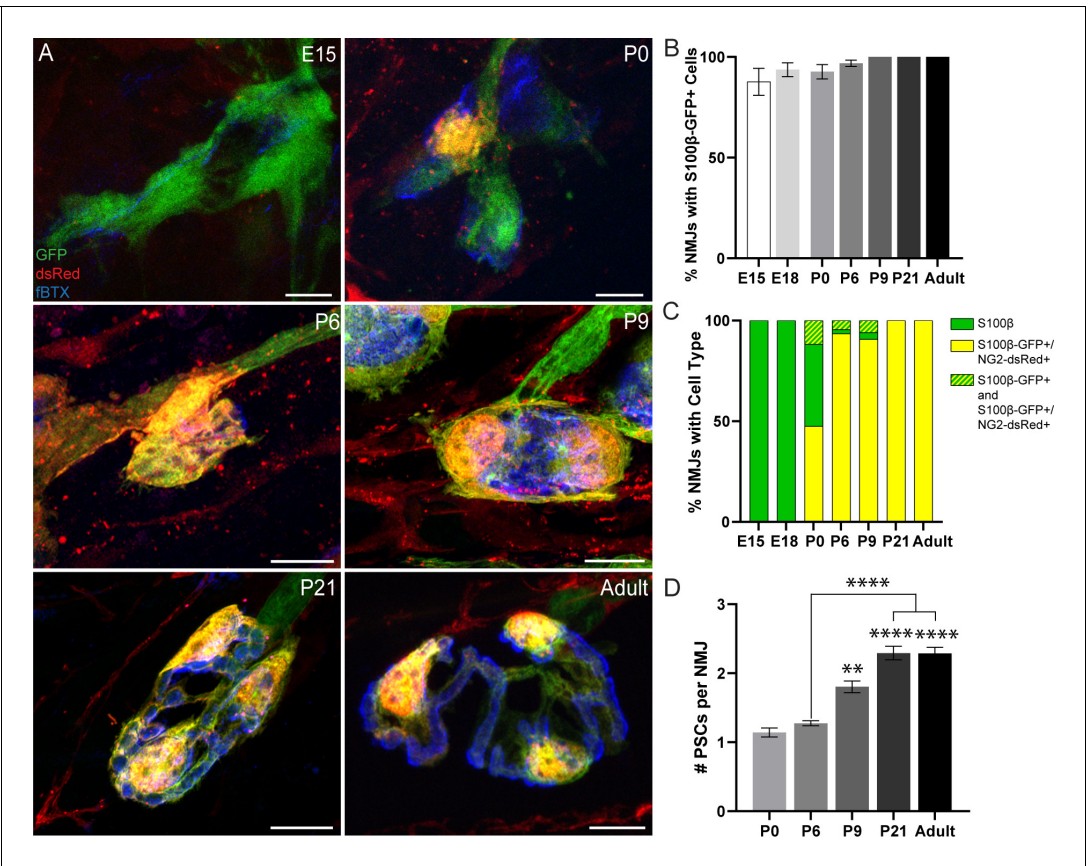

**Figure 3.** Analysis of NG2-dsRed distribution and PSC density during NMJ development in the EDL muscle. (**A**) Representative images of NMJs, identified by nAChR clusters with fBTX (blue), in developing (E15, P0, P6, P9, P21) and adult S100β-GFP (green);NG2-dsRed (red) transgenic EDL. (**B**) The number of NMJs populated by at least one S100β-GFP+ cell increases between the ages of E15 and P9, at which point all observed NMJs have at least one S100β-GFP+ cell. (**C**) Analysis of NMJs that contain at least one single labeled S100β-GFP+ cell (green bar), at least one double labeled S100β-GFP+;NG2-dsRed+ cell (yellow bar) or a combination of single labeled S100β-GFP+ cells and double labeled S100β-GFP+;NG2-dsRed+ cells (green/yellow bar) shows that developing NMJs are exclusively populated by S100β-GFP+ cells in the embryonic stages and are increasingly populated by S100β-GFP+;NG2-dsRed+ cells as the NMJ develops. (**D**) The average number of PSCs per NMJ increases during development. Error bar = standard error of the mean. Scale bar = 10 μm. *=p < 0.05,***=P < 0.001; ****=P < 0.0001. Asterisks represent comparisons with P0 unless otherwise noted.

The online version of this article includes the following figure supplement(s) for figure 3:

**Figure supplement 1.** Color and grayscale images of PSCs in the EDL muscle of (**A**) E15, (**B**) E18, (**C**) P0, (**D**) P6, (**E**) P9, (**F**) P21, and (**G**) adult S100β-GFP;NG2-dsRed mice.

**Figure supplement 2.** Cells at NMJs express NG2 in adults but not at embryonic timepoints.

cells that do not express NG2-dsRed (*Figure 3C*). At post-natal day 0 (P0), however, we observed NG2-dsRed expression in a small subset of S100β-GFP$^+$ cells (*Figure 3A,C*). Notably, the proportion of NMJs with S100β-GFP$^+$;NG2-dsRed$^+$ cells sharply increased between the ages of P0 and P9, coinciding with the period of NMJ maturation in mouse skeletal muscles (*Figure 3C*). By P21, when NMJ maturation in mice is near completion (*Sanes and Lichtman, 1999*), we observed S100β-GFP$^+$;NG2-dsRed$^+$ cells to be exclusively present at NMJs (*Figure 3C*). At this age, the number of S100β-GFP$^+$; NG2-dsRed$^+$ PSCs reached an average of 2.3 per NMJ and this remained unchanged in healthy young adult mice (*Figure 3A,D*). To confirm that dsRed expression from the NG2 promoter denotes the temporal and spatial transcriptional control of the NG2 gene in S100β-GFP;NG2-dsRed mice, we immunostained for NG2 protein. We found NG2 protein present at mature NMJs but not in NMJs of E18 mice with IHC (*Figure 3—figure supplement 2*). Thus, the induced expression of NG2 during the course of NMJ development in Schwann cells located proximally to the NMJ provides further evidence that NG2 is a marker of mature, differentiated S100β$^+$ PSCs. It is possible that PSCs upregulate NG2 during development in order to restrict motor axon growth at the NMJ (*Filous et al., 2014*). Induced NG2 expression during NMJ development along with the constant presence of S100β-GFP$^+$ cells (S100β-GFP$^+$ or S100β-GFP$^+$;NG2-dsRed$^+$) and absence of single labeled NG2-dsRed+ cells at NMJs at every observed developmental time point (*Figure 3B,C*) strongly support previous studies indicating that PSCs originate from Schwann cells (*Lee et al., 2017*).

To gain insights into the rules that govern the distribution of PSCs at NMJs we compared PSC density in EDL, soleus, and diaphragm muscles to determine if PSC density is similar across muscles with varying NMJ sizes, fiber types and functional demands. Here, we observed similar PSC densities in each muscle type (*Figure 4—figure supplement 1*), suggesting that the number of PSCs directly correlates with the size of the NMJ and not the functional characteristics or fiber type composition of the muscles with which they are associated.

We next examined the spatial distribution of PSCs at the NMJ using the Nearest Neighbor (NN) analysis. This analysis measures the linear distance between neighboring cells in order to determine the regularity of spacing (*Wassle and Riemann, 1978*; *Cook, 1996*), quantified using the regularity index. In this analysis, randomly distributed groups of cells yield a nearest neighbor regularity index (NNRI) of 1.91 while those with nonrandom, regularly ordered distributions yield higher NNRI values (*Reese and Keeley, 2015*; *Figure 4A*). We found that the spacing of PSCs yielded high NNRI values and thus maintained ordered, non-random distributions at NMJs in the EDL muscle of adult mice. Moreover, this ordered distribution was maintained regardless of the overall number of PSCs at a given NMJ (*Figure 4B–D*). These observations are in accord with a published study indicating that PSCs occupy distinct territories at adult NMJs (*Brill et al., 2011*). In addition, these data strongly suggest that presynaptic, postsynaptic, and/or PSC-PSC mechanisms of communication dictate the spatial distribution of PSCs.

The ability to distinguish PSCs from all other Schwann cells makes it possible to identify genes that are either preferentially or specifically expressed in PSCs. We deployed fluorescence-activated cell sorting (FACS) to separately isolate double labeled S100β-GFP$^+$;NG2-dsRed$^+$ PSCs, single-labeled S100β-GFP$^+$ Schwann cells, and single-labeled NG2-dsRed$^+$ cells (including α-SMA pericytes and Tuj1$^+$ precursor cells [*Birbrair et al., 2013b*]) from juvenile (P15-P22) S100β-GFP;NG2-dsRed transgenic mice. We then utilized RNA-Sequencing (RNA Seq) to compare the transcriptional profile of PSCs with the other two groups (*Figure 5A*). Light microscopy and expression analysis of GFP and dsRed using quantitative PCR (qPCR) confirmed that only cells of interest were sorted (*Figure 5A–B*). This analysis revealed a unique transcriptional profile for PSCs (*Figure 5C*). Notably, we found 567 genes enriched in PSCs that were not previously recognized to be associated with PSCs, glial cells or synapses (*Supplementary file 1*) using Ingenuity Pathway Analysis (IPA). We also found a number of genes preferentially expressed by PSCs with known roles at synapses (*Mozer and Sandstrom, 2012*; *Fox and Umemori, 2006*; *Rafuse et al., 2000*; *Ranaivoson et al., 2019*; *Shapiro et al., 2007*; *Peng et al., 2010*; *Supplementary file 1*). Providing additional insights about the function of PSCs, IPA revealed synaptogenesis, glutamate receptor, and axon guidance signaling as top canonical pathways under transcriptional regulation (*Figure 5D*).

We next cross-referenced our transcriptomic data with a list of genes compiled from published studies indicating enrichment or functional roles in PSCs (*Young et al., 2005*; *Reynolds and Woolf, 1992*; *Robitaille, 1995*; *Robitaille et al., 1997*; *Rochon et al., 2001*; *Georgiou and Charlton, 1999*; *Trachtenberg and Thompson, 1996*; *Morris et al., 1999*; *Woldeyesus et al., 1999*;

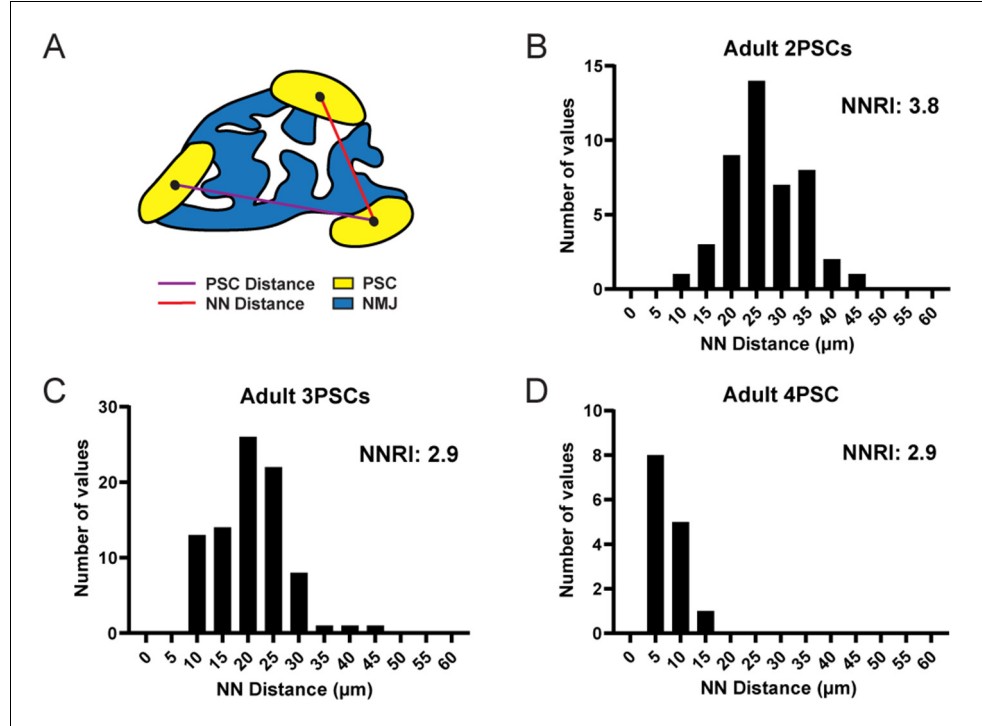

**Figure 4.** PSC distribution at the NMJ is non-random and ordered. (**A**) The nearest neighbor (NN) distance, or the distance between a PSC and the closest neighboring PSC, is represented by the red line. The distance represented by the purple line is not considered for NN analysis. The distribution of NN values (shown in panels B, C, and D) is used to determine the degree of order in PSC distribution, as represented by the nearest neighbor regularity index (NNRI). Distribution patterns with an NNRI value greater than 1.91 are considered to be non-random. (**B–D**) Nearest neighbor distributions of S100-GFP+;NG2-dsRed+ PSCs in adult (P60) EDL muscle show that PSC distributions have orderly patterns of distribution with NNRI > 1.91 regardless of whether they are located at an NMJ with 2 PSCs (**B**), 3 PSCs (**C**), or 4 PSCs (**D**).

The online version of this article includes the following figure supplement(s) for figure 4:

**Figure supplement 1.** The number of PSCs associated with an NMJ varies but PSC density remains constant in the EDL, soleus (SOL), and diaphragm (DIA) muscles of adult mice.

*Riethmacher et al., 1997*; *Personius et al., 2016*; *Park et al., 2017*; *Pinard et al., 2003*; *Descarries et al., 1998*; *Hess et al., 2007*; *Heredia et al., 2018*; *Darabid et al., 2018*; *Musarella et al., 2006*; *De Winter et al., 2006*; *Feng and Ko, 2008*; *Yang et al., 2001*; *Petrov et al., 2014*; *Robitaille et al., 1996*; *Gorlewicz et al., 2009*; *Wright et al., 2009*). This analysis identified 27 genes expressed in isolated S100β-GFP[+];NG2-dsRed[+] PSCs that were previously shown to be associated with PSCs (*Table 1*). These included genes involved in detection and modulation of synaptic activity such as adenosine (*Robitaille, 1995*; *Rochon et al., 2001*), P2Y (*Robitaille, 1995*; *Heredia et al., 2018*; *Darabid et al., 2018*), acetylcholine (*Robitaille et al., 1997*; *Petrov et al., 2014*; *Wright et al., 2009*) and glutamate receptors (*Pinard et al., 2003*), Butyrylcholinesterase (BChE) (*Petrov et al., 2014*), and L-type calcium channels (*Robitaille et al., 1996*). Additionally, genes involved in NMJ development, synaptic pruning, and maintenance including agrin, 2′,3′-cyclic nucleotide 3′ phosphodiesterase (CNP) (*Georgiou and Charlton, 1999*), Erb-b2 receptor tyrosine kinase 2 (Erbb2) (*Trachtenberg and Thompson, 1996*; *Morris et al., 1999*; *Woldeyesus et al., 1999*), Erbb3 (*Trachtenberg and Thompson, 1996*; *Riethmacher et al., 1997*) GRB2-associated protein 1 (Gab1) (*Park et al., 2017*), myelin-associated glycoprotein (MAG) (*Georgiou and Charlton, 1999*), and myelin protein zero (Mpz) (*Georgiou and Charlton, 1999*) were detected in PSCs.

We deployed quantitative PCR (qPCR) to validate preferential expression of select genes in PSCs. To do so, we obtained RNA from S100β-GFP[+];NG2-dsRed[+] PSCs, single-labeled S100β-GFP[+]

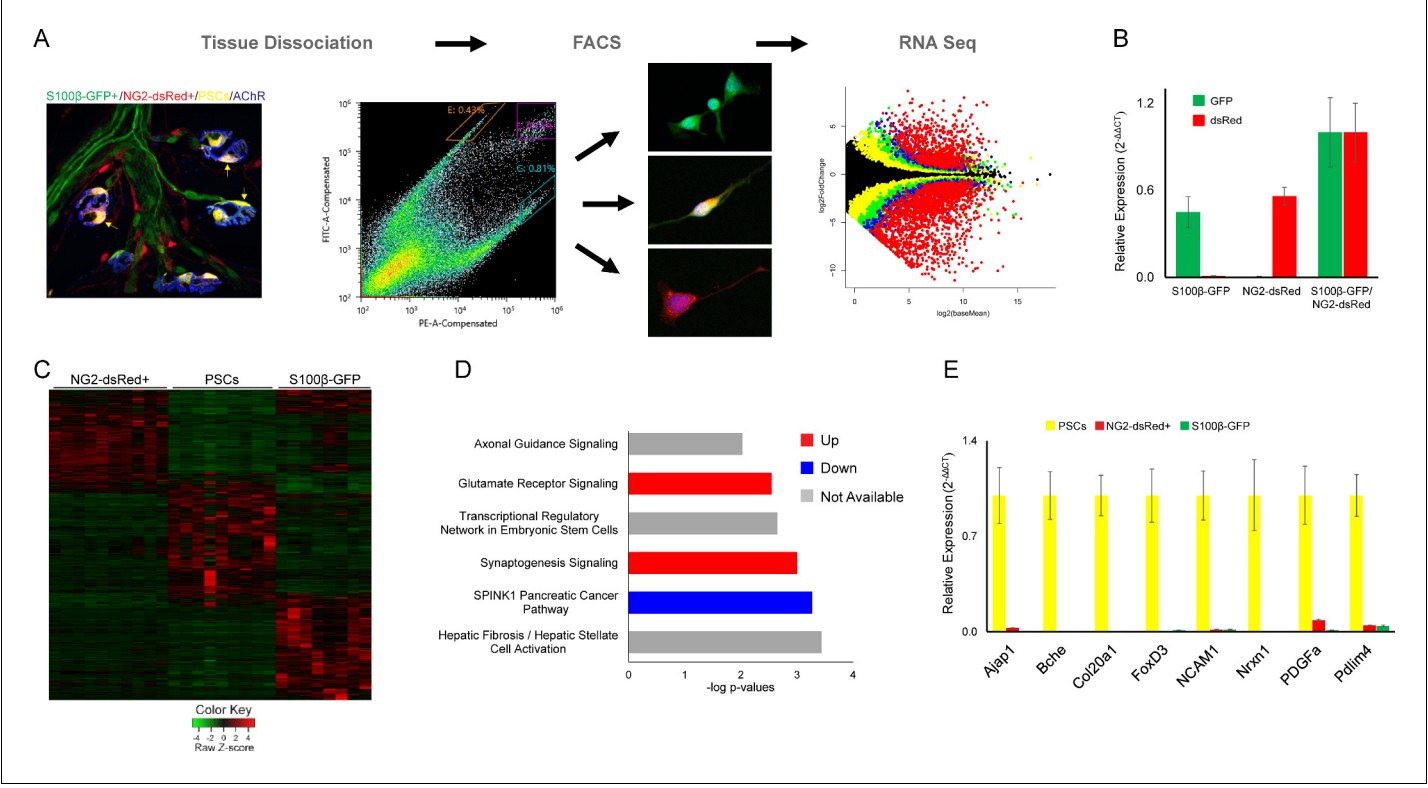

**Figure 5.** Molecular analysis of S100β-GFP+;NG2-dsRed+ PSCs, S100β-GFP+ Schwan cells, and NG2-dsRed+ cells following isolation with FACS. (**A**) Skeletal muscle from juvenile S100B-GFP;NG2-dsRed mice was dissociated and S100-GFP+;NG2-dsRed+ PSCs, S100β-GFP+ Schwan cells, and NG2-dsRed+ cells were sorted by FACS for RNA seq and qPCR. Representative fluorescence intensity gates for sorting of S100β-GFP+, NG2-dsRed+ and S100β-GFP+;NG2-dsRed+ cells are indicated in the scatter plot. GFP (y-axis) and dsRed (x-axis) fluorescence intensities were used to select gates for S100β-GFP+ cells (outlined in orange), NG2-dsRed+ cells (outlined in teal), and double labeled S100β-GFP+;NG2-dsRed+ cells (outlined in purple). Representative images of cells from sorted populations are shown. (**B**) GFP and dsRed qPCR was performed on FACS isolated cells to confirm specificity of sorting gates. (**C**) A heat map of RNA-seq results depicting genes with at least 5 counts and expression differences with a p-value of less than 0.01 between any 2 cell types reveals a distinct transcriptome in S100β-GFP+;NG2-dsRed+ PSCs versus S100β-GFP+ Schwann cells and NG2-dsRed+ cells. (**D**) Synaptogenesis and axon guidance signaling are among the most influential signaling pathways in PSCs according to Ingenuity Pathway Analysis of genes enriched in PSCs versus S100β-GFP+, and NG2-dsRed+ cells. (**E**) qPCR was performed on FACS isolated S100-GFP+;NG2-dsRed+ PSCs, S100β-GFP+ Schwann cells, and NG2-dsRed+ cells to verify mRNA levels of RNA seq identified PSC enriched genes. In each analysis, transcripts were not detected or detected at low levels in S100β-GFP+ Schwann cells and NG2-dsRed+ cells. Error bar = standard error of the mean. Scale bar = 10 μm.

Schwann cells, and single-labeled NG2-dsRed[+] cells isolated using FACS from juvenile S100β-GFP; NG2-dsRed transgenic mice. We examined eight genes identified by RNA seq as being highly enriched in PSCs. This included newly identified genes (Ajap1, Col20a1, FoxD3, Nrxn1, PDGFa, and Pdlim4) and genes previously shown to be enriched (BChE [*Petrov et al., 2014*] and NCAM1 [*Covault and Sanes, 1986*]) in PSCs (*Figure 5E*). Validating RNA-Seq findings, qPCR analysis showed that all eight genes are highly enriched in PSCs compared to all other cell types isolated via FACS (*Figure 5E*). Additionally, immunostaining showed that NG2, a novel PSC-enriched gene identified by RNA-Seq, is concentrated at the NMJ (*Figure 3—figure supplement 2*).

## Discussion

We have discovered a unique combination of molecular markers that allows us to specifically visualize, isolate, interrogate the transcriptome, and potentially alter the molecular composition of PSCs. We have shown that NG2 is specifically expressed by S100β-GFP[+] PSCs but not myelinating S100β-GFP[+] Schwann cells and thus the combined expression of S100β and NG2 is a unique molecular marker of PSCs in skeletal muscle. Providing further evidence that NG2 is a marker of differentiated PSCs, we have demonstrated that Schwann cells induce expression of NG2 shortly after they arrive

**Table 1.** Genes with functional roles in PSCs identified by RNA seq analysis of isolated PSCs.

| Gene | Description | Proposed role | Read count | Log2 change vs NG2-dsRed+ | Log2 change vs S100β-GFP+ | Reference |
|------|-------------|---------------|-----------|-------------|-------------|-----------|
| Adora2a | Adenosine A2a receptor | Detect/modulate synaptic activity | 8.1 | −3.68 | −2.67 | (*Robitaille, 1995*; *Rochon et al., 2001*) |
| Adora2b | Adenosine A2b receptor | Detect/modulate synaptic activity | 9.2 | −3.16 | −4.55 | (*Robitaille, 1995*; *Rochon et al., 2001*) |
| Agrn | Agrin | AChR aggregation | 2049.7 | 1.16 | 2.93 | (*Georgiou and Charlton, 1999*) |
| Bche | Butyrylcholinesterase | Modulate synaptic ACh levels | 7191.0 | 7.89 | 7.21 | (*Trachtenberg and Thompson, 1996*) |
| Cacna1c | L type Calcium channel, alpha 1 c | Detect/modulate synaptic activity | 14.3 | −4.92 | −2.10 | (*Morris et al., 1999*) |
| Cacna1d | L type Calcium channel, alpha 1d | Detect/modulate synaptic activity | 18.4 | −0.42 | −1.49 | (*Morris et al., 1999*) |
| Cd44 | CD44 antigen | Mediates cell-cell interactions | 1249.2 | 0.75 | −1.22 | (*Woldeyesus et al., 1999*) |
| Chrm1 | Muscarinic acetylcholine receptor M1 | Detect/modulate synaptic activity | 14.8 | n.d. | 0.89 | (*Robitaille et al., 1997*; *Riethmacher et al., 1997*) |
| Cnp | 2′,3′-cyclicnucleotide 3′ phosphodiesterase | Anchors axon terminal at NMJ | 2990.2 | 4.23 | 1.66 | (*Personius et al., 2016*) |
| Erbb2 | Erb-b2 receptor tyrosine kinase 2 | Synaptogenesis/ maintenance | 228.9 | 0.84 | 1.37 | (*Park et al., 2017*; *Pinard et al., 2003*; *Descarries et al., 1998*) |
| Erbb3 | Erb-b2 receptor tyrosine kinase 3 | Synaptogenesis/ maintenance | 2471.3 | 7.05 | 4.46 | (*Park et al., 2017*; *Hess et al., 2007*) |
| GAb1 | GRB2-associated protein 1 | Synaptic pruning | 693.8 | 0.31 | 1.57 | (*Heredia et al., 2018*) |
| Grm1 | Glutamate receptor, metabotropic 1 | Detect/modulate synaptic activity | 9.2 | n.d. | 0.80 | (*Darabid et al., 2018*) |
| Grm5 | Glutamate receptor, metabotropic 5 | Detect/modulate synaptic activity | 38.0 | n.d. | 2.84 | (*Darabid et al., 2018*) |
| LNX1 | Ligand of numb-protein X 1 | Regulate NRG1 signaling | 37.5 | −2.29 | −0.70 | (*Peper et al., 1974*) |
| MAG | Myelin-associated glycoprotein | Synaptogenesis/ maintenance | 136.0 | 3.12 | −0.55 | (*Personius et al., 2016*) |
| Mpz | Myelin protein zero | Synaptogenesis/ maintenance | 4590.7 | 2.54 | −0.79 | (*Personius et al., 2016*) |
| Nos2 | Nitric oxide synthase 2, inducible | Synaptogenesis/modulate synaptic activity | 13.4 | −2.91 | −1.28 | (*Musarella et al., 2006*) |
| Nos3 | Nitric oxide synthase 3, endothelial cell | Synaptogenesis/modulate synaptic activity | 48.6 | −2.69 | −0.68 | (*Musarella et al., 2006*) |
| P2ry1 | Purinergic receptor P2Y1 | Detect/modulate synaptic activity, synapse elimination | 144.4 | 0.52 | 2.21 | (*Robitaille, 1995*; *De Winter et al., 2006*; *Feng and Ko, 2008*) |
| P2ry2 | Purinergic receptor P2Y2 | Detect/modulate synaptic activity | 24.0 | −1.55 | −1.04 | (*Robitaille, 1995*) |
| P2ry10b | P2Y receptor family member P2Y10b | Detect/modulate synaptic activity | 10.0 | −1.25 | −3.14 | (*Robitaille, 1995*) |
| P2ry12 | P2Y receptor family member P2Y12 | Detect/modulate synaptic activity | 273.5 | n.d. | 3.70 | (*Robitaille, 1995*) |
| P2ry14 | P2Y receptor family member P2Y14 | Detect/modulate synaptic activity | 13.6 | −3.49 | −2.06 | (*Robitaille, 1995*) |
| S100b | S100 protein beta | Intracellular signaling | 1788.3 | 5.34 | 3.12 | (*Reynolds and Woolf, 1992*) |
| Sema3a | Semaphorin 3a | Detect/modulate synaptic activity | 136.6 | 2.95 | 1.07 | (*Yang et al., 2001*) |
| Tgfb1 | Transforming growth factor, beta 1 | AChR aggregation | 173.2 | −1.08 | −1.90 | (*Petrov et al., 2014*) |

at the NMJ during maturation of the synapse. However, the means by which the induced expression of NG2 is part of a program to establish and/or further specify PSC identity in Schwann cells at the NMJ, through activation of the NG2 promoter, remains to be determined.

We utilized FACS to isolate S100β-GFP$^+$;NG2-dsRed$^+$ PSCs from skeletal muscle to analyze the PSC transcriptome. To our knowledge this is the first time that the transcriptome of PSCs, or any other type of glial cell that associates exclusively with synapses, has been characterized. This analysis reveals expression of a number of genes that have been previously implicated in modulation of synaptic activity, synaptic pruning, and synaptic maintenance by PSCs. We identified a number of novel genes that are highly expressed in PSCs but not Schwann cells or NG2$^+$ cells. These genes have the potential to assist PSC research by serving as molecular markers that can be utilized for PSC-specific genetic manipulations, PSC ablation, and isolation of PSCs for cell culture and molecular analysis. We verified a number of these with qPCR and IHC. This analysis, therefore, reveals a unique gene expression signature that distinguishes PSCs from all other Schwann cells.

While the role of the majority of genes found enriched in PSCs at the neuromuscular synapse remains to be determined, it is worth noting that many have been shown to play key roles in neuronal circuits in the central nervous system and in cell-cell communication. This is the case for NG2 which has been shown to terminate axonal growth in glial scars in the spinal cord (*Filous et al., 2014*). Therefore, it is possible that NG2 is utilized by PSCs to tile, and thus occupy unique territories, and prevent motor axons from developing sprouts that extend beyond the postsynaptic partner. Supporting this possibility, we have found that the NG2 promoter is active in some PSCs at P0 (*Figure 3C*), a time when motor axon nerve endings at NMJs undergo rapid morphological changes (*Sanes and Lichtman, 1999*; *Sanes and Lichtman, 2001*). The progressive activation of the NG2 promoter in PSCs is complete by P9 (*Figure 3C*), which coincides with the elimination of extranumeral axons innervating the same postsynaptic site in mice (*Sanes and Lichtman, 1999*; *Sanes and Lichtman, 2001*). Therefore, PSCs may utilize NG2 to promote the maturation of the presynaptic region and thus the NMJ. Furthermore, PSCs may utilize NG2 to repel each other as they tile during development to occupy unique territories at the NMJ (*Brill et al., 2011*).

With these tools, it is now possible to determine which cellular and molecular determinants are critical for PSC differentiation, maturation, and function at the NMJ. It will also allow us to ascertain the contribution of PSCs to NMJ repair following injury and NMJ degeneration during normal aging and the progression of neuromuscular diseases, such as Amyotrophic Lateral Sclerosis (ALS) and Spinal Muscular Atrophy (SMA). Our strategy of specifically labeling synaptic glia, using a combination of protein markers uniquely expressed in this cell type, may serve as a springboard for unprecedented approaches for studying not only PSC function at the NMJ, but also synapse-associated glia throughout the CNS. Indeed, we have observed subsets of astrocytes in the brain that co-express both S100β and NG2, as has been previously reported in the context of a lineage tracing analysis (*Deloulme et al., 2004*). Future studies will determine the generality of our approach in discerning the functional roles of synaptic glia in the development, maintenance, and function of select synapses.

## Materials and methods

### Key resources table

| Reagent type (species) or resource | Designation | Source or reference | Identifiers | Additional information |
|---|---|---|---|---|
| Genetic reagent (*M. musculus*) | *S100b-GFP* | PMID:15590915 | MGI:3588512 | Dr. Wesley Thompson (Texas A and M) |
| Genetic reagent (*M. musculus*) | *NG2-dsRed* | PMID:18045844 | MGI:3796063 | Dr. Akiko Nishiyama (University of Connecticut) |
| Genetic reagent (*M. musculus*) | *SOD1$^{G93A}$* | PMID:8209258 | MGI:2183719 | Dr. Deng (Northwestern University) |

*Continued on next page*

*Continued*

| Reagent type (species) or resource | Designation | Source or reference | Identifiers | Additional information |
|---|---|---|---|---|
| Antibody | Guinea pig polyclonal anti-NG2 | PMID:19058188 | Antibody Registry: AB_2572299 | 1:250 |
| Antibody | Alexa Fluor-488 goat polycolonal anti guinea pig | Invitrogen | RRID:AB_2534117 | 1:1000 |
| Antibody | Alexa Fluor-488 goat polyclonal anti rabbit | Invitrogen | Catalog# A-11008 | 1:1000 |
| Software, algorithm | Ingenuity Pathway Analysis | Qiagen | RRID:SCR_008117 | |
| Software, algorithm | GraphPad Prism | GraphPad | RRID:SCR_002798 | |
| Software, algorithm | R | The R Project for Statistical Computing | RRID:SCR_001905 | |
| Software, algorithm | ImageJ | ImageJ | RRID:SCR_003070 | |
| Software, algorithm | Bio-Rad CFX Manager | Bio-Rad | RRID:SCR_017251 | |
| Commercial assay or kit | PicoPure RNA Isolation Kit | ThermoFisher | Catalog#KIT0204 | |
| Commercial assay or kit | iScript cDNA synthesis kit | Bio-Rad | Catalog#1708891 | |
| Commercial assay or kit | SsoAdvanced PreAmp Supermix | Bio-Rad | Cataolog#1725160 | |
| Commercial assay or kit | iTAQ Univeral SYBR Green Supermix | Bio-Rad | Catalog#1725121 | |
| Chemical compound, drug | Alexa Fluor-555 alpha-bungarotoxin | Invitrogen | Catalog#B35451 | |
| Chemical compound, drug | DAPI | ThermoFisher | Catalog#D1306 | |

## Mice

SOD1$^{G93A98}$ (*Gurney et al., 1994*), S100β-GFP (B6;D2-Tg(S100b-EGFP)1Wjt/J) (*Zuo et al., 2004*) and NG2-dsRed mice (Tg(Cspg4-DsRed.T1)1Akik/J) (*Zhu et al., 2008*) were obtained from Jackson Labs (Bar Harbor, ME). S100β-GFP and NG2-dsRed mice were crossed to generate S100β-GFP;NG2-dsRed mice. Offspring were genotyped using Zeiss LSM900 to check for fluorescent labels. SOD1$^{G93A}$ mice were crossed with S100β-GFP;NG2-dsRed mice to generate S100β-GFP;NG2-dsRed;SOD1$^{G93A}$ mice. Postnatal mice older than 9 days of age were anesthetized and immediately perfused with 4% paraformaldehyde (PFA) overnight. Pups were anesthetized by isoflurane and euthanized by cervical dislocation prior to muscle dissociation. Adult mice were anesthetized using $CO_2$ and then perfused transcardially with 10 ml of 0.1 M PBS, followed by 25 ml of ice-cold 4% PFA in 0.1 M PBS (pH 7.4). All experiments were carried out under NIH guidelines and animal protocols approved by the Brown University and Virginia Tech Institutional Animal Care and Use Committee.

## Fibular nerve crush

Adult S100β-GFP;NG2-dsRed mice were anesthetized with a mixture of ketamine (100 mg/kg) and xylazine (10 mg/kg) delivered intraperitoneally. The fibular nerve was crushed at its intersection with

the lateral tendon of the gastrocnemius muscle using fine forceps, as described previously (*Dalkin et al., 2016*). Mice were monitored for 2 hr following surgery and administered buprenorphine (0.05-.010 mg/kg) at 12 hr intervals during recovery.

## Immunohistochemistry and NMJ visualization

For NG2 immunohistochemistry (IHC), muscles were incubated in blocking buffer (5% lamb serum, 3% BSA, 0.5% Triton X-100 in PBS) at room temperature for 2 hr, incubated with anti-NG2 antibody (courtesy of Dr. Dwight Bergles) diluted at 1:250 in blocking buffer overnight at 4°C, washed 3 times with 0.1M PBS for 5 min. Muscles were then incubated with 1:1000 Alexa Fluor-488 conjugated anti-rabbit or guinea pig antibody (A-11008, Invitrogen, Carlsbad, CA) and 1:1000 Alexa Fluor-555 conjugated α-bungarotoxin (fBTX; Invitrogen, B35451) in blocking buffer for 2 hr at room temperature and washed 3 times with 0.1M PBS for 5 min. For all other NMJ visualization, muscles were incubated in Alexa Fluor-647 conjugated α-bungarotoxin (fBTX; Invitrogen, B35450) at 1:1000 and 4′,6-Diamidino-2-Phenylindole, Dihydrochloride (DAPI; D1306, ThermoFisher, Waltham, MA) at 1:1000 in 0.1M PBS at 4°C overnight. Muscles were then washed with 0.1M PBS 3 times for 5 min each. Muscles were whole mounted using Vectashield (H-1000, Vector Labs, Burlingame, CA) and 24 × 50–10.5 cover glass (ThermoFisher).

## Confocal microscopy of PSCs and NMJs

All images were taken with a Zeiss LSM700, Zeiss LSM 710, or Zeiss LSM 900 confocal light microscope (Carl Zeiss, Jena, Germany) with a 20 × air objective (0.8 numerical aperture), 40 × oil immersion objective (1.3 numerical aperture), or 63 × oil immersion objective (1.4 numerical aperture) using the Zeiss Zen Black software. Optical slices within the z-stack were taken at 1.00 μm or 2.00 μm intervals. High resolution images were acquired using the Zeiss LSM 900 with Airyscan under the 63 × oil immersion objective in super resolution mode. Optical slices within the z-stack were 0.13 μm with a frame size of 2210 × 2210 pixels. Images were collapsed into a two-dimensional maximum intensity projection for analysis.

## Image analysis

### NMJ size

To quantify the area of NMJs, the area of the region occupied by nicotinic acetylcholine receptors (nAChRs, labeled by fBTX, was measured using ImageJ software. At least 100 nAChRs were analyzed for number of fragments, individual nicotinic acetylcholine receptor (nAChR) clusters, from each muscle to represent an individual mouse. At least 3 animals per age group were analyzed to generate the represented data.

### Cells associated with NMJs

Cell bodies were visualized via GFP and/or dsRed signal, and were confirmed as cell bodies by the presence of a DAPI+ nucleus. The area of each cell body was measured by tracing the outline of the entire cell body using the freehand tool in ImageJ. To quantify the number of cells associated with NMJs, the number of cell bodies directly adjacent to each NMJ was counted. Every cell that overlapped with or directly abutted the fBTX signal was considered adjacent to the NMJ. At least 3 animals per age group were analyzed to generate the represented data. Cells were examined in at least 100 NMJs from each muscle to represent an individual mouse.

### Spacing of PSCs at NMJs

NMJs were identified via fBTX signal. PSCs were identified by the colocalization of GFP, dsRed, and DAPI signal in addition to their location at NMJs. The area of each PSC and NMJ was measured. The linear distance from the center of each PSC soma to the center of the nearest PSC soma at a single NMJ was measured. The distances were then separated into 5 μm bins and plotted in a histogram. All linear measurements were made using the line tool in the ImageJ software. At least 100 NMJs were analyzed from each muscle to represent an individual mouse.

## Muscle dissociation and fluorescence activated cell sorting

Diaphragm, pectoralis, forelimb and hindlimb muscles were collected from P15-P21 S100β-GFP; NG2-dsRed mice. After removal of connective tissue and fat, muscles were cut into 5 mm$^2$ pieces with forceps and digested in 2 mg/mL collagenase II (Worthington Chemicals, Lakewood, NJ) for 1 hr at 37°C. Muscles were further dissociated by mechanical trituration in Dulbecco's modified eagle medium (Life Technologies, Carlsbad, CA) containing 10% horse serum (Life Technologies) and passed through a 40 µm filter to generate a single cell suspension. Excess debris was removed from the suspension by centrifugation in 4% BSA followed by a second centrifugation in 40% Optiprep solution (Sigma-Aldrich, St. Louis, MO) from which the interphase was collected. Cells were diluted in FACS buffer containing 1 mM EDTA, 25 mM Hepes, 1% heat inactivated fetal bovine serum (Life Technologies), in $Ca^{2+}/Mg^{2+}$ free 1X Dulbecco's phosphate buffered saline (Life Technologies). FACS was performed with a Sony SH800 Cell Sorter (Sony Biotechnology, San Jose, CA). Representative fluorescence intensity gates for sorting of S100-GFP$^+$, NG2-dsRed$^+$ and S100-GFP$^+$;NG2-dsRed$^+$ cells are provided in *Figure 5*. Purity of the sorted cell population was confirmed by visual inspection of sorted cells using an epifluorescence microscope and with dsRed and GFP qPCR (*Figure 5A,B*). A single mouse was used for each replicate and an average of 7500 cells per replicate were collected for each cell group.

## RNA-seq and qPCR

RNA was isolated from S100β-GFP$^+$, NG2-dsRed$^+$, or S100β-GFP$^+$/NG2-dsRed$^+$ cells following FACS with the PicoPure RNA Isolation Kit (ThermoFisher). The maximum number of cells that could be collected by FACS following dissociation of muscles collected from one mouse was used as a single replicate. On average, a single replicate consisted of 7,500 cells. RNA seq was performed by Genewiz on 12 replicates per cell type. Following sequencing, data were trimmed for both adaptor and quality using a combination of ea-utils and Btrim (*Aronesty, 2013*; *Kong, 2011*). Sequencing reads were aligned to the genome using HiSat2 (*Kim et al., 2019*) and counted via HTSeq (*Anders et al., 2015*). QC summary statistics were examined to identify any problematic samples (e.g. total read counts, quality and base composition profiles (+/- trimming), raw fastq formatted data files, aligned files (bam and text file containing sample alignment statistics), and count files (HTSeq text files). Following successful alignment, mRNA differential expression were determined using contrasts of and tested for significance using the Benjamini-Hochberg corrected Wald Test in the R-package DESeq2 (*Love et al., 2014*). Failed samples were identified by visual inspection of pairs plots and removed from further analysis resulting in the following number of replicates for each cell type: NG2-dsRed$^+$, 10; S100β-GFP$^+$, 7; NG2-dsRed$^+$;S100β-GFP$^+$, 9. Functional and pathway analysis was performed using Ingenuity Pathway Analysis (QIAGEN Inc, https://www.qiagenbio-informatics.com/products/ingenuity-pathway-analysis). Confirmation of expression of genes identified by RNA-seq was performed on 6 additional replicates of each cell type using quantitative reverse transcriptase PCR (qPCR). Reverse transcription was performed with iScript (Bio-Rad, Hercules, CA) and was followed by a preamplification PCR step with SsoAdvanced PreAmp Supermix (Bio-Rad) prior to qPCR using iTAQ SYBR Green and a CFX Connect Real Time PCR System (Bio-Rad). Relative expression was normalized to 18S using the $2^{-\Delta\Delta CT}$ method. The primers used for both preamplification and qPCR are listed in *Supplementary file 2*.

### Statistics

An unpaired t-test or one-way ANOVA with Bonferroni post hoc analysis were used for statistical evaluation. Data are expressed as the mean ± standard error of the mean, and $p < 0.05$ was considered statistically significant. The number of replicates is as follows: RNA seq, 7–10; qPCR, 6; comparisons between EDL, soleus, and diaphragm, 1; comparison between SOD1$^{G93A}$ and healthy adult muscle, 1; all other analyses, 3. Statistical analyses were performed using GraphPad Prism8 and R. Data values and p-values are reported within the text.

## Acknowledgements

We thank members of the Valdez laboratory for providing helpful comments throughout the course of this project.

## Additional information

### Funding

| Funder | Grant reference number | Author |
|---|---|---|
| National Institutes of Health | R01AG055545 | Gregorio Valdez |
| National Institutes of Health | R56AG051501 | Gregorio Valdez |
| National Institutes of Health | R21NS106313 | Gregorio Valdez |

The funders had no role in study design, data collection and interpretation, or the decision to submit the work for publication.

### Author contributions

Ryan Castro, Thomas Taetzsch, Formal analysis, Investigation, Methodology, Writing - original draft, Writing - review and editing; Sydney K Vaughan, Data curation, Formal analysis, Writing - original draft, Writing - review and editing; Kerilyn Godbe, Formal analysis, Writing - review and editing; John Chappell, Resources, Writing - review and editing; Robert E Settlage, Formal analysis, Methodology, Writing - original draft, Writing - review and editing; Gregorio Valdez, Conceptualization, Resources, Data curation, Formal analysis, Supervision, Funding acquisition, Investigation, Methodology, Writing - original draft, Writing - review and editing

### Author ORCIDs

Ryan Castro https://orcid.org/0000-0002-2316-8039
Thomas Taetzsch http://orcid.org/0000-0003-3257-1142
Sydney K Vaughan https://orcid.org/0000-0002-3427-4654
Robert E Settlage http://orcid.org/0000-0002-1354-7609
Gregorio Valdez https://orcid.org/0000-0002-0375-4532

### Ethics

Animal experimentation: All experiments were carried out under NIH guidelines and animal protocols approved by the Brown University (IACUC# 19-05-0013) and Virginia Tech (IACUC# 18-148 and 18-176) Institutional Animal Care and Use Committee.

### Decision letter and Author response

Decision letter https://doi.org/10.7554/eLife.56935.sa1
Author response https://doi.org/10.7554/eLife.56935.sa2

## Additional files

### Supplementary files

- Supplementary file 1. Genes with highly enriched expression in perisynaptic Schwann cells.
- Supplementary file 2. Primers used for cDNA preamplification and qPCR.
- Transparent reporting form

### Data availability

RNA-Seq data has been deposited in NCBI GEO and all other data is available in the main text or the supplementary materials. The GEO accession number for this dataset is GSE152774.

The following dataset was generated:

| Author(s) | Year | Dataset title | Dataset URL | Database and Identifier |
|---|---|---|---|---|
| Castro R, Taetzsch T, Vaughan SK, Godbe K, Chappell | 2020 | Synaptic Schwann cells: specific labeling reveals unique cellular and molecular features | https://www.ncbi.nlm.nih.gov/geo/query/acc.cgi?acc=GSE152774 | NCBI Gene Expression Omnibus, GSE152774 |

J, Settlage RE, Val-
dez G

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
