## [Decision Letter]

**Acceptance summary:**

Perisynaptic Schwann cells (PSCs) at the neuromuscular junction (NMJ) play essential roles in synaptic function, formation, maintenance, and repair. However, the molecular characteristics specific to PSCs are largely unknown. In this manuscript, Castro et al., have adopted a novel approach of combining S100B and NG2 labels to reveal a specific probe for PSCs and identify several associated novel genes. Their new discoveries advance molecular understanding of synapse-glia interactions at the NMJ in health and disease, and provide a foundation for further exploration.

**Decision letter after peer review:**

Thank you for submitting your article "Specific labeling reveals unique cellular and molecular features of synaptic Schwann cells" for consideration by *eLife*. Your article has been reviewed by three peer reviewers, and the evaluation has been overseen by a Reviewing Editor and Gary Westbrook as the Senior Editor. The following individual involved in review of your submission has agreed to reveal their identity: Chien-Ping Ko (Reviewer #2). The reviewers have discussed the reviews with one another and the Reviewing Editor has drafted this decision to help you prepare a revised submission.

We would like to draw your attention to changes in our revision policy that we have made in response to COVID-19 (https://elifesciences.org/articles/57162). Specifically, when editors judge that a submitted work as a whole belongs in *eLife* but that some conclusions require a modest amount of additional new data, as they do with your paper, we are asking that the manuscript be revised to either limit claims to those supported by data in hand, or to explicitly state that the relevant conclusions require additional supporting data. Our expectation is that the authors will eventually carry out the additional experiments and report on how they affect the relevant conclusions either in a preprint on bioRxiv or medRxiv, or if appropriate, as a Research Advance in *eLife*, either of which would be linked to the original paper.

Summary:

The manuscript by Castro et al. describes the use of combined S100b and NG2 to identify perisynaptic (terminal) Schwann cells at the neuromuscular junction. The role of PSCs in the development, function, degeneration and regeneration of the NMJ is well established, but their careful study has been limited by a lack of molecular markers. This study convincingly demonstrates that combined expression of S100b (labeling all Schwann cells) and NG2, labeling only relatively mature terminal Schwann cells (as well as other cell types in muscle) can be combined to study, and more importantly isolate, PSCs. Some data presented largely reproduce previous findings regarding PSCs, but this information serves to validate the use of S100b/NG2 to identify this population. Given the long-standing interest in identifying specific markers of PSCs to allow their isolation and manipulation, these findings are important for the field. However, a few items would improve the manuscript and data presentation. as guided by the Essential revisions section below.

Essential revisions:

1) Address each of the key comments (below) in the revised paper, especially regarding the potential limitations and uncertainties of the new marker, as raised by both reviewers.

2) Transcriptional Profiling of PSCs

– More could be done to validate the top DE genes using RNA scope and in situ hybridization. That is relatively straightforward technically if this is possible during covid19 period. For example, the authors provide only one IHC for validation beyond the qPCR and it is not clear why an adherens junction (Ajap1) would show diffuse cytoplasmic localization, so more will need to be done to validate this antibody and show that immunoreactivity is indeed AJAP1 (in addition to validating other DE genes). The authors are referred to the above paragraph regarding *eLife* policy during COVID-19 for our expectations.

– Are there other NG2^+^ cells in muscle and if so, this needs to be considered in the interpretation of the current RNA-seq data from this population?

– The presentation of the RNA-seq data could be formatted to be a little more useful to help rank candidates of interest, including read count and fold change (similar format to Table 1). See comments reviewer 2.

Revisions expected in follow-up work:

– Additional validation and spatial mapping of more DE expressed genes (sm FISH with combinations of markers).

Reviewer #1:

Comments have been incorporated into the editorial comments above.

Reviewer #2:

Perisynaptic Schwann cells (PSCs) at the neuromuscular junction (NMJ) play essential roles in synaptic function, formation, maintenance, and repair. However, the molecular characteristics specific to PSCs are largely unknown. In this manuscript, Castro et al. have adopted a novel approach of combining S100B and NG2 labels to reveal a specific probe for PSCs and identify several associated novel genes. The manuscript is well-written and comprehensive. Their findings are truly exciting, compelling, and immensely significant, as the field desperately needs specific probes for PSCs. Their new discoveries would advance molecular understanding of synapse-glia interactions at the NMJ in health and disease.

1) Introduction, last paragraph and Results, first paragraph. The authors may like to comment on whether they are the first group to show NG2 at PSCs and/or how they came up with the idea of combining both S100B and NG2 for their approach to identify PSCs. It would also be useful to comment/speculate the potential function of NG2 and other genes such as Ajap1 in PSCs.

2) Results, first paragraph, Figure 1F. The authors use the AChR fragmentation index to assess if NMJ deterioration occurs in their transgenic mouse line. It would be more sensitive and informative by also labeling the presynaptic nerve terminals to check if there is denervation. In addition, it is desirable to examine the quantal contents to confirm if indeed there are no functional deficits at the NMJ in their transgenic mice.

3) It is unclear about the statement, "we failed to find.… throughout the course of embryonic and postnatal development." But NG2 is expressed at least partially starting from P0 (Figure 2C). The authors may also like to comment/speculate on the reason/significance for NG2 expression only for mature differentiated PSCs.

4) The authors used FACS to isolate S100B-GFP^+^;NG2-dsRed^+^ PSCs from skeletal muscles to analyze the PSC transcriptome. However, it appears that NG2 expresses only in PSCs after birth and in mature PSCs (Figure 2C, Figure 2—figure supplement 1). It is also not known whether denervation after nerve injury and subsequent repair would alter the expression of NG2. If NG2 is downregulated or absent in denervated PSCs, their new probes for PSCs and associated genes might not be able to address the role of PSCs after nerve injury. It would be desirable to perform a relatively straightforward experiment by simply examining NG2 staining after nerve injury. This additional experiment would broaden the applications of using their transgenic line to address a key role of PSCs in synaptic repair in future studies. Otherwise, their approaches would be somewhat limited to studying only the mature PSCs in intact muscles.

Reviewer #3:

The paper by Castro et al. describes the use of combined S100b and NG2 to identify perisynaptic (terminal) Schwann cells at the neuromuscular junction. Besides the identification of this marker combination, the most significant results presented are the transcriptional profile of these cells isolated by flow cytometry from muscle. Given the long-standing interest in identifying specific markers of PSCs to allow their isolation and manipulation, these findings are important for the field. The image quality in the paper is excellent, and the quantification and statistics are rigorous and convincing. A few items would further improve the manuscript and data presentation.

The authors should comment on what the other NG2^+^ cells in muscle may be (recognizing there may be multiple cell types). This would help in future studies attempting to use a similar intersectional approach to study PSCs and also in the interpretation of the current RNA-seq data from this population (indeed, the -seq may help with the identification of these cell types).

The list of genes in Supplementary file 1 that also show enrichment in the S100b^+^/NG2^+^ cells versus either single marker is a potentially very useful resource for the field. As such, it should include some additional information to help rank candidates of interest, including read count and fold change (similar format to Table 1, without references and proposed functions).

There is a lot of future potential to be based on this work (response to denervation/axotomy, status and gene expression changes in disease, genetic ablation, etc.). Addition of such studies would strengthen this paper, but also require significant time and effort and exactly which of these studies would be most important and most informative is hard to predict. Therefore, the intrinsic value of publishing the identification of these markers and the list of PSC-enriched genes is noted as it will allow the field to subsequently pursue these additional experiments.

---

## [Author Response]

Essential revisions:1) Address each of the key comments (below) in the revised paper, especially regarding the potential limitations and uncertainties of the new marker, as raised by both reviewers.2) Transcriptional Profiling of PSCs– More could be done to validate the top DE genes using RNA scope and in situ hybridization. That is relatively straightforward technically if this is possible during covid19 period. For example, the authors provide only one IHC for validation beyond the qPCR and it is not clear why an adherens junction (Ajap1) would show diffuse cytoplasmic localization, so more will need to be done to validate this antibody and show that immunoreactivity is indeed AJAP1 (in addition to validating other DE genes). The authors are referred to the above paragraph regarding eLife policy during COVID-19 for our expectations.

We thank the reviewer for this comment, and agree that further validation of differentially expressed genes would strengthen this dataset. However, our lab does not currently possess the proper tools and equipment to perform in situ hybridization and/or RNA scope. Obtaining and setting up this equipment would take considerable time, and would require the purchase of new materials and equipment. It is unlikely that this will be possible in the immediate future due to COVID-19 restrictions, which for example continues to keeps us away from our lab and will make it difficult to purchase additional items not deemed necessary. Please do note that we do intend to obtain the equipment necessary to begin performing these and other important techniques for assessing PSCs in our lab once we resume “normal” research activities and as funds become available. However, we hope that our current validation using immunohistochemistry and qPCR will be sufficient at this time. Given that it will not be possible to validate our Ajap1 antibody with Ajap1 knockout tissue, we have removed the Ajap1 IHC results from the manuscript (changes made to Figure 5 and Results, last paragraph).

– Are there other NG2^+^ cells in muscle and if so, this needs to be considered in the interpretation of the current RNA-seq data from this population?

The reviewer has brought attention to an area of the manuscript that is currently lacking in sufficient detail. We have updated the text to include a better account of the types of cells that express NG2 in muscles based on published work by Birbrair A, et al., 2013a, b (changes made to Results, first and eighth paragraphs). We have also updated the text to reflect the fact that the collection of NG2-dsRed^+^ cells from muscles for RNA-seq represents a heterogeneous population of cell-types.

– The presentation of the RNA-seq data could be formatted to be a little more useful to help rank candidates of interest, including read count and fold change (similar format to Table 1). See comments reviewer 2.

We have modified the presentation of our RNA-seq data in Supplementary file 1 to include read count and fold change. Our entire RNA-seq data-set has been deposited in the NCBI-GEO database as well, so all of this information will be readily available to the reader.

Revisions expected in follow-up work:– Additional validation and spatial mapping of more DE expressed genes (sm FISH with combinations of markers).Reviewer #1:Comments have been incorporated into the editorial comments above.Reviewer #2:[…] 1) Introduction, last paragraph and Results, first paragraph. The authors may like to comment on whether they are the first group to show NG2 at PSCs and/or how they came up with the idea of combining both S100B and NG2 for their approach to identify PSCs. It would also be useful to comment/speculate the potential function of NG2 and other genes such as Ajap1 in PSCs.

We thank the reviewer for pointing out this important omission. This manuscript concerns the discovery of new molecular markers, and we failed to explain in sufficient detail how that discovery was made. We have updated the text to rectify this mistake (changes made to Results, first paragraph). We have also expanded the text to include a more detailed discussion of the possible roles of NG2 in NMJ development (changes made to Discussion, third paragraph).

2) Results, first paragraph, Figure 1F. The authors use the AChR fragmentation index to assess if NMJ deterioration occurs in their transgenic mouse line. It would be more sensitive and informative by also labeling the presynaptic nerve terminals to check if there is denervation. In addition, it is desirable to examine the quantal contents to confirm if indeed there are no functional deficits at the NMJ in their transgenic mice.

We agree with the reviewer that analysis of presynaptic nerve terminals and quantal content would contribute to our analysis here, and we cannot discount the possibility that there are currently unidentified negative consequences of the co-expression of GFP and dsRed in PSCs. Unfortunately, performing these additional analyses would take a great deal of time and resources, and would be difficult to complete in a timely manner as long as COVID-19 restrictions persist. In this revision, we have added that we see no discernible differences between general features of S100B-GFP^+^ animals and NG2dsRed^+^/S100B-GFP^+^ animals, such as overall health or basic motor capacity (changes made to Results, second paragraph). We have also added a sentence that discusses this particular limitation of the study. Additionally, we see no changes in PSC number (Figure 1E) between NG2-dsRed^+^ and NG2-dsRed^-^ animals.

3) It is unclear about the statement, "we failed to find.… throughout the course of embryonic and postnatal development." But NG2 is expressed at least partially starting from P0 (Figure 2C). The authors may also like to comment/speculate on the reason/significance for NG2 expression only for mature differentiated PSCs.

We agree with the reviewer that we were unclear in the quoted section of the text. We have updated the text to clarify when NG2 appears in PSCs. Additionally, we included text describing the possible function of NG2 at the NMJ (changes made to Results, fourth paragraph).

4) The authors used FACS to isolate S100B-GFP^+^;NG2-dsRed^+^ PSCs from skeletal muscles to analyze the PSC transcriptome. However, it appears that NG2 expresses only in PSCs after birth and in mature PSCs (Figure 2C, Figure 2—figure supplement 1). It is also not known whether denervation after nerve injury and subsequent repair would alter the expression of NG2. If NG2 is downregulated or absent in denervated PSCs, their new probes for PSCs and associated genes might not be able to address the role of PSCs after nerve injury. It would be desirable to perform a relatively straightforward experiment by simply examining NG2 staining after nerve injury. This additional experiment would broaden the applications of using their transgenic line to address a key role of PSCs in synaptic repair in future studies. Otherwise, their approaches would be somewhat limited to studying only the mature PSCs in intact muscles.

We agree with the reviewer and have added additional data and information that demonstrate the wide-spread utility of these markers to study PSCs at NMJs under various stressors (Figure 2E-H). This includes data showing that PSCs continue to be the only cells expressing both GFP and dsRed (proxies for S100B and NG2) after crushing innervating motor axons and during the symptomatic stage of ALS in a mouse model of the disease characterized by significant destruction of NMJs (changes made to Results, third paragraph). These data strongly indicate that these markers will be sufficient to study PSCs in diseases, injury and potentially during aging.

Reviewer #3:The paper by Castro et al. describes the use of combined S100b and NG2 to identify perisynaptic (terminal) Schwann cells at the neuromuscular junction. Besides the identification of this marker combination, the most significant results presented are the transcriptional profile of these cells isolated by flow cytometry from muscle. Given the long-standing interest in identifying specific markers of PSCs to allow their isolation and manipulation, these findings are important for the field. The image quality in the paper is excellent, and the quantification and statistics are rigorous and convincing. A few items would further improve the manuscript and data presentation.The authors should comment on what the other NG2^+^ cells in muscle may be (recognizing there may be multiple cell types). This would help in future studies attempting to use a similar intersectional approach to study PSCs and also in the interpretation of the current RNA-seq data from this population (indeed, the RNA-seq may help with the identification of these cell types).

We agree, see response to Essential revisions.

The list of genes in Supplementary file 1 that also show enrichment in the S100b^+^/NG2^+^ cells versus either single marker is a potentially very useful resource for the field. As such, it should include some additional information to help rank candidates of interest, including read count and fold change (similar format to Table 1, without references and proposed functions).

We agree, see response to Essential revisions.

There is a lot of future potential to be based on this work (response to denervation/axotomy, status and gene expression changes in disease, genetic ablation, etc.). Addition of such studies would strengthen this paper, but also require significant time and effort and exactly which of these studies would be most important and most informative is hard to predict. Therefore, the intrinsic value of publishing the identification of these markers and the list of PSC-enriched genes is noted as it will allow the field to subsequently pursue these additional experiments.